# HESS Opinions: A Planetary Boundary on Freshwater Use is Misleading

Maik Heistermann [1]

[1] Institute of Earth and Environmental Science, University of Potsdam, 14476, Germany

*Correspondence to*: Maik Heistermann (maik.heistermann@uni-potsdam.de)

**Abstract:** In 2009, a group of prominent earth scientists introduced the "planetary boundaries" (PB) framework: They suggested nine global control variables, and defined corresponding *"thresholds which, if crossed, could generate unacceptable environmental change"*. The concept builds on systems theory, and views Earth as a complex adaptive system in which anthropogenic disturbances may trigger non-linear, abrupt and irreversible changes at the global scale, and *"push*

*the Earth system outside the stable environmental state of the Holocene"*. While the idea has been remarkably successful in both science and policy circles, it has also raised fundamental concerns, as the majority of suggested processes and their corresponding planetary boundaries do not operate at the global scale, and thus apparently lack the potential to trigger abrupt planetary changes.

This paper picks up the debate with specific regard to the planetary boundary on "global freshwater use". While the bio-physical impacts of excessive water consumption are typically confined to the river basin scale, the PB proponents argue that water-induced environmental disasters could build up to planetary scale feedbacks and system failures. So far, however, no evidence has been presented to corroborate that hypothesis. Furthermore, no coherent approach has been presented to what extent a planetary threshold value could reflect the risk of regional environmental disaster. To be sure, the PB framework

was revised in 2015, extending the planetary freshwater boundary with a set of basin-level boundaries inferred from environmental water flow assumptions. Yet, no new evidence was presented, neither with respect to the ability of those basin-level boundaries to reflect the risk of regional regime shifts, nor with regard to a potential mechanism linking river basins to the planetary scale.

So while the idea of a planetary boundary on freshwater use appears intriguing, the line of arguments presented so far remains speculative and implicatory. As long as Earth system science does not present compelling evidence, the exercise of assigning actual numbers to such a boundary is arbitrary, premature and misleading. Taken as a basis for water-related policy and management decisions, though, the idea transforms from misleading to dangerous, as it implies that we can globally offset water-related environmental impacts. A planetary boundary on freshwater use should thus be disapproved and actively

refuted by the hydrological and water resources community.

# 1 The Planetary Boundaries Framework

In 2009, a group of prominent scientists lead by Johan Rockström introduced the "Planetary Boundaries"[1] (PB) framework (Rockström et al., 2009a, 2009b). They identified nine Earth system processes – *climate change, rate of biodiversity loss, interference with the nitrogen and phosphorus cycles, stratospheric ozone depletion, ocean acidification, global freshwater use, land use change, chemical pollution, and atmospheric aerosol loading* –, each of which is represented by a control variable. Accordingly, planetary boundaries are defined as *"thresholds [of these control variables] which, if crossed, could generate unacceptable environmental change"*. Moving outside this *"safe operating space for humanity"* may be *"deleterious or even catastrophic for human well-being."*

At its heart, the PB framework builds on systems theory. It views Earth as a complex adaptive system in which anthropogenic disturbances may trigger non-linear, abrupt and irreversible changes at the global scale, and *"push the Earth system outside the stable environmental state of the Holocene"*. Furthermore, Rockström and colleagues refer to concepts such as limits-to-growth (Meadows et al., 2004), safe minimum standards (Crowards, 1998), the precautionary principle (Raffensperger and Tickner, 1999), and tolerable windows (Petschel-Held et al., 1999).

The success of the PB framework has been remarkable in both scientific and policy arenas. Since 2009, the two original papers together have been cited more than 2000 times in scientific journals tracked by the Thomsen Reuters's Web of Science, and are still gaining traction. The PB concept has been embraced by the United Nations High-Level Panel on Global Sustainability (2012), and non-governmental organizations such as the World Wildlife Fund for Nature (2016). It was included in the Global Environment Outlook 5 (United Nations Environment Programme, 2012), and underpins a reform proposal for global environmental institutions by the Earth System Governance Project (Biermann et al. 2012). In April 2017, a large international conference on "Making the Planetary Boundary Concept Work" is hosted by the German government, namely the *Federal Ministry for the Environment, Nature Conservation, Building and Nuclear Safety* and the *Federal Environment Agency* (Umweltbundesamt).

The PB framework is founded on the assumption that transgressing any of the planetary boundaries may induce irreversible changes at the global scale. In this paper, I will argue that the definition of a corresponding planetary boundary on freshwater use is not only scientifically weak, but misleading and potentially dangerous if operationalised in a policy context. It should thus be disapproved and actively refuted by the hydrological and water resources community.

---

[1] not to be confused with the meteorological term „planetary boundary layer", i.e. the lowest part of the atmosphere

## 2 Previous Debate

The PB framework was quickly picked up in the scientific discourse. Critical commentaries, however, mostly called for a revision of the actual numbers (e.g. Molden, 2009; Destouni et al., 2013; Jaramillo and Destouni, 2015). Compared to the widespread endorsement of the PB framework, fundamental criticism has been scarce. Only few authors insisted that the majority of suggested processes and their corresponding planetary boundaries do not operate at the global scale, and thus do not have the potential to trigger abrupt planetary changes. Accordingly, Lewis (2012) concluded that *"there is no need for all the world's countries to enter protracted legal discussions on aggregate boundaries: those affected by regional problems should work among themselves to solve them. Global negotiations should focus on managing the clear planetary boundaries of climate change and ocean acidification [...]"*. Nordhaus et al. (2012) argued that *"six of the planetary boundaries [...] do not have planetary biophysical thresholds [...] and operate on local to regional, not global, levels."* They warn that "*global limits may risk misleading local and regional policy choices*".

In order to safeguard against such concerns, the original paper by Rockström et al. (2009a) had already distinguished between *"boundaries that are directly related to sharp continental or planetary thresholds [...], and boundaries based on 'slow' planetary processes with no current evidence of planetary scale threshold behavior [...]"*. Interestingly, the admitted lack of evidence did not keep the authors from defining the corresponding planetary thresholds, hypothesizing that these *"may arise at the local and regional scales, which become a global concern at the aggregate level."*

Resulting from a continued discourse, Steffen et al. (2015) published a revised version of the PB framework in *Science*, updating most of the boundary estimates, but also trying to consider some of the more fundamental criticism. Most importantly, the revision introduces *"a two-tier approach for several of the boundaries to account for regional-level heterogeneity"* (see next section). Furthermore, Steffen et al. (2015) repeatedly insist on the necessity to define planetary boundaries for *all* of the processes included in the framework, e.g. claiming that

> *"[...] not all Earth-system processes included in the PB approach have singular thresholds at the global/continental/ocean basin level. Nevertheless, it is important that boundaries be established for these processes [...] [such as freshwater use]. Placing boundaries for these processes is more difficult [...] but is nevertheless important for maintaining the resilience of the Earth system as a whole."*

In the following section, both the *original* and the *revised* version of the framework will be discussed with specific regard to the *planetary boundary on freshwater use*.

So far, the hydrological community has remained quite silent in the controversy about the scientific justification of the PB framework. The discourse did not take place in "traditional" hydrology or water resources journals. Freshwater use is, however, at the heart of hydrological science and water resources management, and it is about time for the community to take a stand towards a corresponding planetary boundary.

## 3 The Planetary Boundary on Freshwater Use

Rockström et al. (2009a, 2009b) suggested that global freshwater consumption (from rivers and groundwater bodies) by humans must not exceed 4000 km$^3$ per year, while the current level of that control variable was estimated at a level of 2600 km$^3$. These figures remained essentially unchanged in the update by Steffen et al. (2015). Accordingly, the current level of

human freshwater appropriation is still considered to be "safe" (see Fig. 1).

While such a conclusion might be doubted by those billions of people already exposed to water scarcity, I neither intend to question the estimate of the boundary, nor the estimate of the control variable's current value, but the concept of a planetary freshwater boundary itself.

Generally, we expect environmental impacts of consumptive freshwater use to be confined to the river basin scale. Exceptions are e.g. with interbasin water transfer schemes (Zhuang, 2016), and system collapses such as the Aral Sea Disaster, for which the biophysical and socioeconomic impacts could, in fact, be felt at a regional scale beyond the watershed (Micklin, 2007). It is that kind of regional regime shift that the PB proponents most likely had in mind. Yet, they

do not corroborate how such a collapse could push the entire Earth system away from its Holocene state. Such a planetary feedback would only be conceivable through mechanisms in the climate system. And, truly, terrestrial moisture recycling is a mechanism that can link regions far apart from each other – beyond watersheds, and potentially across continents (van der Ent et al., 2010). It replenishes, through terrestrial evapotranspiration, the moisture flux that is directed from the oceans into the continents, and thus sustains downwind rainfall. Hence, there is growing concern that e.g. large-scale deforestation might

fundamentally disrupt moisture recycling (e.g. Boers et al., 2017). However, the dynamic role of local coupling and changes in the atmospheric circulation are yet to be understood. Accordingly, Goessling and Reick (2011) warned that *"moisture recycling estimates cannot consistently be used as reliable indicators for the sensitivity of precipitation to modified land-evaporation"*.

But while there is at least credible evidence that deforestation could disrupt regional water cycles, the role of consumptive freshwater use, e.g. by irrigation, remains largely unclear: a number of studies suggest that irrigation intensifies terrestrial soil moisture recycling, and thus increases downwind precipitation (e.g. DeAngelis et al., 2010; Puma and Cook, 2010; Jódar et al., 2010; Harding and Snyder, 2012; Zou et al., 2014; Alter et al., 2015). Others suggest that irrigation affects local

rainfall, too: irrigation-induced surface cooling could increase local atmospheric stability, and thus reduce local rainfall (Lee et al., 2009; Guimberteau et al., 2012; Im et al., 2014; Tuinenburg et al., 2014). Or, irrigation could increase convective available potential energy and precipitable water, and thus increase local rainfall (Mahalov et al., 2016). Irrigation-induced changes in local precipitation are mostly tied to changes in large-scale moisture convergence, and thus to changes of

precipitation elsewhere. Besides, local and downwind effects can occur simultaneously (Pei et al., 2016). Im and Eltahier (2014) even detected, in a simulation experiment, a constellation in which irrigation increased rainfall (by 100 %) and runoff (by 50 %) in the Niger River basin *upstream* from the irrigation location.

What all of these studies demonstrate, together, is that the complex interaction of different atmospheric and surface

processes is yet poorly understood. What *none* of these studies demonstrates, though, is how freshwater use would cause the collapse of regional or continental hydrological cycles. Accordingly, neither Rockström et al. (2009a) nor Steffen et al. (2015) have presented evidence to support their claim that *"water-induced thresholds at the continental or planetary scale may be crossed as a result of aggregate sub-system impacts at local (e.g., river basin) or regional (e.g., monsoon system) scales caused both by changes in water resource use and climate change-induced shifts in the hydrological cycle"*. Instead,

the line of argument remains implicatory when it refers, in the section on global freshwater use, to studies on wet-to-dry state shifts of the Sahel zone (Scheffer et al. 2001; Foley et al. 2003) and the "savannization" of the Amazon (Oyama and Nobre, 2003) – none of which considers freshwater consumption as a driving force.

But even if we assumed, for a moment, the validity of the hypothesis: that human freshwater use could trigger regional scale

shifts which in turn would built up to planetary scale feedbacks and system failures: How could a *planetary* boundary on freshwater use reflect such *regional* thresholds? Obviously, it could not.

This realisation motivated the extension of the PB framework by Steffen et al. (2015) in order to *"capture the importance of subglobal change for the functioning of the Earth system"*. It is crucial to understand that this extension does not aim at

merely representing the spatial heterogeneity of environmental stress, but to detect regional environmental stress that could feedback to the planetary scale: *"We emphasize that our subglobal-level focus is based on the necessity to consider this level to understand the functioning of the Earth system as a whole."*

Steffen and colleagues maintained the planetary freshwater boundary on "consumptive blue water use ($km^3$/year)" at a level

of 4000-6000 $km^3$/year, but they added *"a basin-scale boundary for the maximum rate of blue water withdrawal along rivers, based on the amount of water required in the river system to avoid regime shifts in the functioning of flow-dependent ecosystems."* That control variable is based on the concept of environmental water flows, and had originally been proposed by Gerten et al. (2013) in the PB context. Yet again, no evidence is presented that those thresholds actually serve their designated purpose within the PB framework (*"... to understand the functioning of the Earth system as a whole."*). It is not

even exemplarily verified that exceeding such a basin level threshold could trigger regional regime shifts, and least of all how such regional shifts could build up to a planetary feedback. What remains, between the lines, is the mere implication that reservations towards a single "planetary freshwater boundary" have been taken care of. Still, the actual relationship between the "planetary freshwater boundary" and its basin-level counterpart remains vague: Gerten et al. (2013) had

originally upscaled the basin-level freshwater boundaries to a single planetary freshwater boundary of 1100 - 4500 km$^3$ / year. But just as Gerten et al. (2013) had not elaborated on the need to aggregate their estimate to a single global number, Steffen et al. (2015) did not explicate why they did *not* aggregate the basin level boundaries, but rather stuck with the original planetary boundary value.

Summing up, the revised framework maintains global freshwater use as a planetary boundary, prominently displayed in the main figure of the *Science* paper (Fig. 1), and widely disseminated thereafter. According to that figure, humanity is still "in the green" with regard to freshwater consumption. In order to rebut concerns regarding such a planetary boundary, Steffen et al. (2015) suggested an additional boundary on freshwater withdrawal at the basin scale, the role of which in the entire PB framework remains vague, and which is not supported by any new evidence.

## 4 The Context of Global Water Governance

For a broader view on the issue, it should be noted that the idea of a planetary freshwater boundary is, intentionally or not, well in line with other concepts implying that water resources could and should be globally managed. The most prominent of these concepts is the *Water Footprint*, originally suggested by Hoekstra and Hung (2002). The Water Footprint is defined

*"as the total volume of freshwater used to produce the goods and services consumed by the individual or community [such as a nation or humanity]"* (Hoekstra et al., 2011). From early on, it had been a deliberate decision to make the Water Footprint a measure of water *use* only, and not to consider water availability at the location in which the water is actually used. The rationale behind that decision is that water is considered a globally scarce resource. Accordingly, reducing the Water Footprint is generally seen as a desirable outcome irrespective of local water scarcity. That notion has attracted fierce

criticism (Gawel and Bernsen, 2013; Perry, 2014; Wichelns, 2010; Wichelns, 2011 - to name only a few), the consequences of which are, however, *nil*: since 2006, more than 700 articles have been published on the topic of "Water Footprint", with a total citation count of more than 10,000 (still exponentially increasing). Similar to the PB framework, the concept has been embraced by the scientific community, by national governments, by non-governmental organisations such as the World Wildlife Fund For Nature, and also by multinational corporations such as Coca Cola, Nestlé, or Unilever. The proponents

continue to emphasize that *"[...] reducing water footprints in water-stressed catchments displays a limited perspective on the question of what is globally sustainable [...]"*, and that *"the world's [...] freshwater resources are accessible from anywhere through trade in water-intensive commodities"* (Hoekstra and Mekonnen, 2012). The latter notion is one of the key arguments in the call for "global water governance" (Hoekstra, 2011; Vörösmarty et al., 2015), based on the hypothesis

that virtual water trade makes water a global resource that, through trade interventions, could be arbitrarily redistributed across basins. This view, however, has been repeatedly refuted (e.g. Gawel and Bernsen, 2013; Wichelns 2015): while a globalised trade in fact "transports" substantial volumes of virtual water across the globe (Dalin et al., 2012), policy and management choices should be made by the affected stakeholders instead of being imposed by whatever water-related global

trade mechanism.

It would surely be worthwhile, in another paper, to provide a comprehensive synopsis of the water footprint debate and its links to the controversy on the planetary freshwater boundary. Both have entirely different motivations: the freshwater PB is about critical environmental limits to water use while the water footprint is about the actual magnitudes of that use. Yet, both

share the implication that the world requires global water governance - a global regulation of water consumption. Meanwhile, the same world still awaits the first evidence that either of the two concepts has yet provided any useful guidance to those actors on the ground who struggle for the sustainable management of an increasingly scarce resource.

## 5 Conclusions

The PB concept can be viewed from two perspectives: first, as a scientific framework that is built on systems theory, and second, as a guide towards sustainable development and resource management. The planetary boundary on freshwater use fails in both regards.

From a *scientific perspective*, the existence of such a boundary is mere speculation. The proponents argue that local

freshwater consumption could lead to regional system collapses which could in turn build up (*"across scales"*) to irreversible state shifts at the global level. While the thought itself is intriguing, the line of arguments presented so far remains implicatory. And as long as Earth system science does not present compelling evidence, the exercise of assigning actual numbers to such a boundary is arbitrary, premature and misleading. It is misleading in multiple respects: it pretends a level of understanding that is nonexistent, and it suggests that reducing global water consumption mitigates regional water

issues.

Still, one might argue that it is worth sacrificing some scientific rigour if the framework could at least pragmatically guide us towards *sustainable water resources management*. In that regard, however, it fails even more obviously. Today, there is no robust evidence how water management in one basin would physically affect other basins across the world, and it is

unsettling to see that scientists and policymakers are starting to take that narrative seriously nonetheless. As Lewis (2012) put it, the idea is *"politically seductive"*. However, taken as a basis for water-related policy and management decisions, it is misleading and potentially dangerous. It suggests that we can globally offset water-related environmental impacts, a notion that defies both common sense and hydrological science. The potential consequences of such reasoning are exemplified in

Fig. 2. While I believe that such ideas were not originally intended by Rockström and colleagues, a planetary boundary on freshwater use remains a point-blank invitation to promote ideas such as "water neutrality" or "water offsetting".

Admittedly, the precautionary principle (Raffensperger and Tickner, 1999) can always be considered as a safeguard against an alleged "lack of scientific evidence". But before kicking off another debate as to whether the hypothesis of a "planetary freshwater boundary" qualifies for a "minimal threshold of plausibility" (van den Belt, 2003), we might find that stressing the precautionary principle simply misses the point: The impacts of water scarcity on human welfare are already obvious, felt every day by the very people living in water scarce regions. Contemplating on the applicability of the precautionary principle to the issue of freshwater use might, to those people, appear like a discussion from a parallel universe.

Given the intensity of that criticism, it appears legitimate to inquire about the original motives to include a process such as freshwater use in a framework on planetary boundaries. Maybe the sheer significance of water – as a key component of the earth system and as a sustainable development challenge – made it agreeable to tacitly ignore hydrological fundamentals? While the key literature on planetary boundaries does not answer that question, the interactive discussion related to this opinion article (http://www.hydrol-earth-syst-sci-discuss.net/hess-2017-112/#discussion) sheds some light on the motives. There, Johan Rockström argues that the planetary boundary on freshwater use actually *"has nothing to do with human water use [but] with the maximum level of shifts in the global hydrological cycle [...]"*.

So, is the present controversy just a matter of terminology? Indeed, Rockström's surprising notion calls, at least, for a fundamental revision of the freshwater PB, starting with a definition that is explicit and transparent with regard to the underlying mechanism: if, for example, the disruption of terrestrial moisture recycling was considered critical, that notion should be clearly reflected by the definition of any water-related boundary. Still, it would be a long way from there to convey the required quantitative evidence whether and at which point that process might *"push the Earth system outside the stable environmental state of the Holocene"*. Until then, the PB community should withstand the temptation to (expert) guess numbers, and instead be as explicit about the fundamental knowledge gap as it should be about the underlying mechanism. Fig. 3 is a mere example how such explicitness could be conceived.

But while the need to fundamentally revise the planetary boundaries framework is obvious, the complete package has gained so much traction that it already appears to be beyond fundamental scrutiny. Still, the hydrological community should not just give in. Instead, we should actively engage in refuting and pushing back the misconception that a global threshold on freshwater use can have any meaningful policy implications, and stop giving scientific credibility to that framework until substantial evidence is presented.

## Acknowledgements

I would like to thank those who participated in the interactive discussion of this manuscript, namely Murugesu Sivapalan, Christof Lorenz, Fernando Jaramillo, Hubert H. G. Savenije, Dieter Gerten, Chris Perry, Johan Rockström, and an anonymous referee. In my opinion, that interactive discussion provides lots of additional insights, and I sincerely recommend

it to any interested reader: http://www.hydrol-earth-syst-sci-discuss.net/hess-2017-112/#discussion.

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

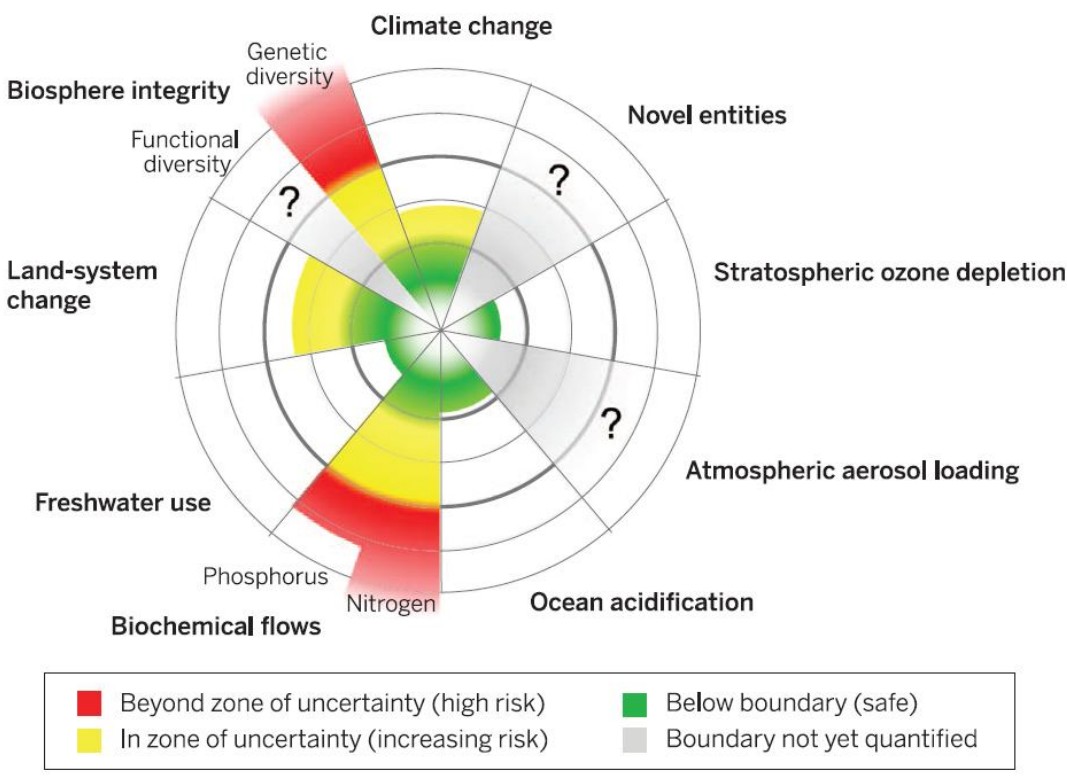

**Figure 1: According to the latest revision of the PB framework, humanity is still "in the green" with regard to freshwater use; from Steffen et al. (2015), reprinted with permission from AAAS.**

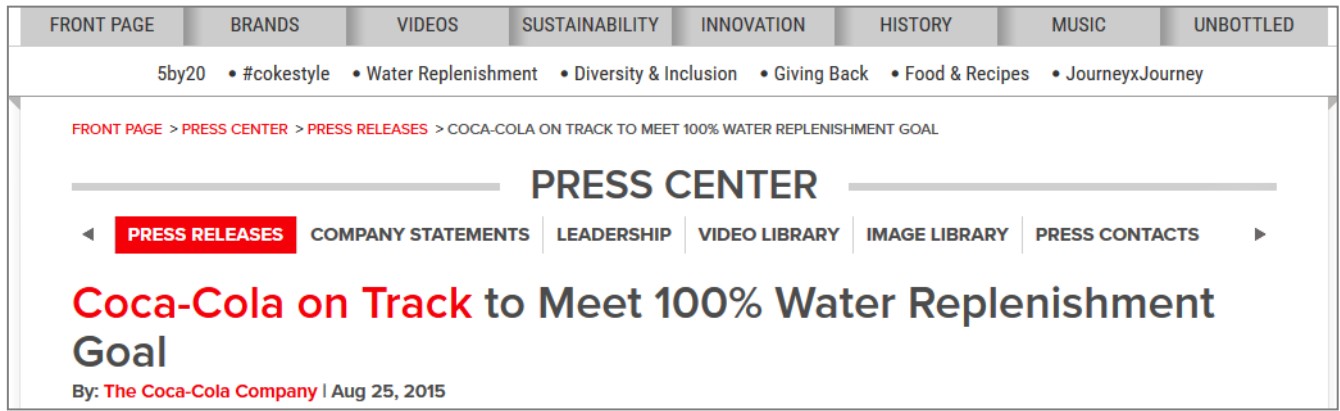

**Figure 2: Screen shot of an online press release from August 25, 2015 (The Coca Cola Company, 2015). In this press release, the Coca Cola Company claims being close to "water neutral". For this purpose, water use related to the production process is offset ("replenished") by conservation efforts around the globe. The press release was quickly and mostly uncritically picked up by various media channels, e.g. the New York Times online edition.**

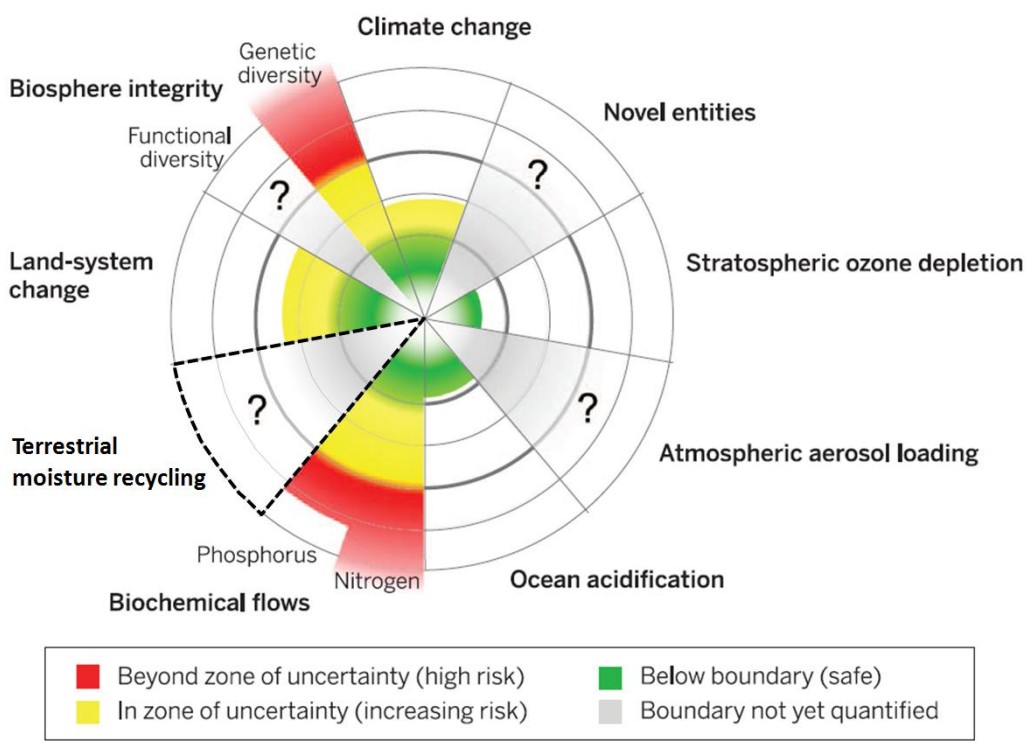

**Figure 3: An example how a water-related planetary boundary could be explicit with regard to both the mechanism of interest and our lack in its understanding; modified from Steffen et al. (2015), with permission from AAAS.**