# Peer review of "HESS Opinions: A Planetary Boundary on Freshwater Use is Misleading"

_Hydrology and Earth System Sciences, 2017_

## Short Comment (SC1) · 12 Mar 2017

I warmly welcome this opinion paper in HESS and appreciate the courage of Dr Heistermann in voicing these opinions. It must be especially difficult for him, given also that he is surrounded by a large community of fellow scientists doing PB research. I also suspect that Dr Heistermann represents the views of a section of the hydrologic community, admittedly small, who have followed the literature on the subject. A much larger section of the community, however, is either not aware of the PB idea, or is somewhat ambivalent on it, because it does not have any bearing on the work they do.

The author evaluated the concept of the planetary boundary on freshwater use from two perspectives: as a scientific framework that is built on systems theory, and as a guide towards sustainable development and resource management. He argues that

the PB on freshwater use fails in both regards. I have to say that I agree with him on both counts.

Having said that, I do believe that the notion of a safe operating safe does make sense at the local or watershed scale, i.e., human freshwater use must not exceed some threshold at the local scale beyond which the environment is irretrievably degraded. However, this threshold limit is different in different places, governed by local climate and hydrology, and the nature of the local human water use (e.g., for drinking water, for food production etc.). Sustainable management requires humans to manage this threshold not only through water conservation (e.g., efficiency of water use), but also through change of the local economies, through exchange of water with neighboring places which may be more well-endowed with water resources, and through trade of commodities produced using the water. Given the heterogeneity, and the nature of these interactions between places, any estimate of the planetary boundary at the small scale cannot simply be aggregated to the global scale: the strong non-linearities make it non-additive, and hence if at all there exists a planetary scale boundary on freshwater use, it is more of an emergent quantity.

Secondly, as the author correctly argues, the PB community has also not demonstrated that once this global threshold is exceeded (if such a thing does exist), something dramatic is going to happen to the Earth system. It is still to be demonstrated using a much more detailed model of the Earth system (e.g., global hydrology model that incorporates heterogeneous human impacts, including real and virtual water transfers at all scales) that the crossing of the threshold will produce a fundament or qualitative transformation of the freshwater crisis (no less than a quantum leap). For example, it must lead to a total breakdown in water security, food shortages, environmental degradation etc. leading to regional or global wars etc. The current definitions of the PB do not go anywhere near this, and I am not in a position to imagine that such an eventuality will come to pass. Please note that if one were to make the same argument for phosphorus (which is currently mined) I can believe it, but not for water.

[Figure]

I also agree with the author, for similar reasons, that accurate quantification of the PB for freshwater use is not going to lead to sustainable water management. However, the same notion applied at the local or watershed scale can indeed lead to sustainable management. The reason is that there is not a world government, nor a world management authority for water resources. Water management is highly devolved (and needs to be) to watershed, regional and national scales. There is a big difference between managing $CO_2$ globally and managing water globally: $CO_2$ in the atmosphere is a constant across the world, and is measurable and so can be easily monitored. Even in this case, we know how hard it has been for governments to agree to do something about managing $CO_2$ and the global warming that results. With water, the problem hits closer to him, and yet global water management is much harder – I would say impossible. Exactly as the author says, it is even dangerous: if I am part of a small community in a corner of China or India, I would not want the management of my local water resources to be controlled by somebody sitting in Washington DC or Geneva. It could be the worst form of colonialism, and bring out the worst of globalization, even if practiced by otherwise well-intentioned people.

My last comment is on the author's call for hydrologists to resist the moves towards imposing the PB idea through instruments of the United Nations. I agree with the author that there is a sense that it is being pushed down our throats. Still I am somewhat ambivalent– I am not sure how to approach this. One can take the attitude the author adopts, which is for the hydrologic community to organize together to resist this move. This is hard, because apart from a small section of the community, most hydrologists are not in tune with this debate. Besides the hydrology community is already divided between pure hydrologists and water resource systems people, sitting in their ivory towers, and so we cannot easily mobilize the community towards something like this.

An alternative approach could be to adopt some of the concepts in PB research that we can live with and do it our own way. Like I said earlier, the concept of the safe operating space for humanity at the small scale does make sense to me, and we can

organize ourselves around this idea, and perform analyses on the factors that govern this safe operating space (climate, culture, socio-economic conditions etc.), and how they change with increasing spatial scale. This is fundamental science and we have a lot of experience dealing with these issues in hydrology. We have an ongoing decadal initiative called, Panta Rhei: Change in Hydrology and Society, spearheaded by the International Association of Hydrological Sciences. Like-minded people could come together to form a grassroots working group to work on this topic: IAHS and Panta Rhei will strongly welcome this. My main request is for the author to indicate how we as a community can organize ourselves and engage in research activities, and leadership that can present another way forward. Just complaining, and writing a few critical commentaries cannot be the solution.

For too long we hydrologists have sat in our ivory towers, and have tended to often forget that we are, in the final analysis, water scientists and, if so, water and humanity cannot be separated. Past isolation from addressing these broader issues have not served us well. The scientific community behind PB are very powerful, well connected at the UN and other global entities. They will do what they will do, they are not necessarily answerable to hydrologists (even if sometimes they speak as if they are hydrologists), yet they are helping to place the water agenda in the top echelons of global diplomacy and governments. If we play our cards well, it might be good for us in the long term. In other words, I would suggest that we adopt the idea that "if you can't beat them, join them". When we do, we should still be true to our science, and only do what is right by our science.

This is a welcome commentary: but commentary is not enough, the author needs to present an alternative way forward

---

## Referee Comment (RC1) · Anonymous Referee #1 · 22 Mar 2017

Comments for the Authors

The manuscript is well written and provocative. The author questions whether or not the discussion of planetary boundaries in the literature is supported sufficiently by scientific inquiry. This is an important and timely piece of work, which will stimulate further discussion among academics.

Specific Comments

Some of the discussion is somewhat too colloquial for a scholarly journal article. For example, one should avoid using contractions such as can't and don't. Also, it is not clear that the situation of two authors not co-publishing a paper is pertinent. It might be better to avoid putting that perspective forward. In general, the manuscript would benefit from an effort to improve the scholarly level of the writing quality.

---

## Author Comment (AC1) · 24 Mar 2017

I would like to thank Prof. Sivapalan for his comments which I perceive, in general, as positive.

I will not reiterate the many points on which I agree with Prof. Sivapalan. Instead, I will only respond on the key issue I disagree with.

We tend to hesitate in appreciating criticism that is expressed without constructively formulating an alternative approach. The present opinion paper lacks such a constructive approach, and Prof. Sivapalan accordingly demands the outline of an "alternative way forward". I disagree with this demand, and in the following, I'll explain my reasons.

I think we do not have to go as far as Popper's scientific theory, according to which

scientific progress is based on falsification. I would, however, like to insist that rejecting the planetary boundary on freshwater use is justified *in itself*. In the manuscript under discussion, I express the view that the idea of a freshwater PB is not only scientifically flawed, but could seriously mislead policy choices. To put it in simple terms: I am convinced that we're better off without the freshwater PB - even if it was not replaced by an "alternative way forward". I'd prefer *not* to obliterate that main message of the paper by embarking on a fundamental discussion of water management frameworks.

Having said that, however, I am optimistic that we might agree that alternatives already exist. Putting behind us the notion that regional water stress can trigger global disaster, we are still faced with the challenge of sustainable water resources management, and there is a lot of reasonable guidance and conceptual framing towards this challenge. Thus, I do not see the necessity to "board" the idea of a "safe operating space" and somehow transform or adopt it to the local or basin level. Instead, I would like to leave the issue by referring to one of the most enjoyable reads on water resources management I've had in the past years: In his paper on the "ABCDE+F" framework, C. Perry (2013) lays out the components he considers as crucial for an effective water management - "effective" meaning that "the outcome [...] is consistent with the declarations of policy made at the relevant level of administration." According to Perry, "effective" does not necessarily mean "good" - a term that expresses certain preferences which *might* be transient: "[...] privatization, for example, was rarely mentioned 20 years ago, was universally promoted 10 years ago, and now appears to be in decline."

In the "ABCDE+F" framework, hydrology comes into play in the "A" part which stands for "Accounting" (for the available resources). This accounting needs to be based on our best knowledge, combining local and regional expertise with state-of-the-art monitoring and simulation technology - and an explicit onsideration of uncertainty and knowledge gaps. This is a crucial part, and well worth the attention of IAHS and Panta Rhei. Considering though, that there are also the B, C, D and E components, hydrologists should not be tempted to overrate their role in water recources management. Still, the fact that we are just reiterating - in a hydrological journal, in the 21st century! - that the basin scale is the fundamental scale to address water resources issues, puts a spotlight on the current debate. Obviously, there's still some work to do. This work is tedious and complex, and not nearly as attractive as declaring global numbers.

Summing up, I can very well understand Prof Sivapalan's demand for perspective. But again, I have to emphasize that I prefer to keep the focus of this opinion paper on rejecting the freshwater PB. And at any cost, I want to avoid the notion that the idea of a freshwater PB could be somehow tuned or tweaked towards something useful.

Of course, Prof. Sivapalan is right that the PB community is not answerable to hydrologists - and certainly not to a small-time scientist like me. Thus, my first and foremost ambition is to just put my opinion on record. That is one step, and one step only. Step by step, we might see if the debate leads us anywhere useful. Certainly, Prof. Sivapalan's comment was already a very valuable part of that debate, and I would like to thank him again for his effort.

References:

Perry, C. (2013): ABCDE+F: a framework for thinking about water resources management, Water International, 38(1), 95–107.

СЗ

---

## Author Comment (AC2) · 24 Mar 2017

I would like to thank the referee for his or her feedback.

Admittedly, the language of the article deviates from what we'd generally expect from a scientific paper. In my view, however, the opinion format justifies certain stylistic liberties. Then again, I have to agree with the referee that the language should, in general, aim at a neutral (or scholarly) style, even if the message might be considered provocative. I am certainly willing to reexamine the manuscript accordingly.

I also agree that the statement on co-authorship (p. 6, ll. 20-21) could be considered as "not to the point". In my view, it illustrates the fact that the concepts of the Water Footprint and the Planetary Boundaries have quite different histories, but, in my opinion, independently arrive at similar conclusions with regard to the need for global water

governance. I understand, though, that it could be also be seen as a "trivia" piece of information which I would thus be willing to drop in a revised manuscript.
* * *

---

## Author Comment (AC3) · 24 Mar 2017

I would like to thank Dr Lorenz for his substantial comments.

He concludes that *"besides the conceptual shortcomings [...], the concept of the planetary boundary on freshwater must be questioned due to insufficient data quality and quantity."* In particular, Dr Lorenz highlights the fundamental gaps that exist in our quantitive understanding of regional and global water cycles. If I rephrase him correctly, these gaps preclude the usage of global models such as LPMml for a meaningful appraisal of freshwater resources or guidance on water management, and he presents a large body of literature (from his own research) to support these statements.

First of all, I would like to respond that I consider the concept of a planetary boundary on freshwater use as flawed - irrespective of our confidence in the underlying procedure on water balance modelling. Even if we *could* close regional and global water budgets, a global threshold would still be meaningless, as should become clear from the manuscript under discussion. This is why I did deliberately not address the issue of data and model uncertainty in the manuscript.

Having said that, I appreciate Dr Lorenz' comments very much! More generally speaking, many (not only global) assessments and modelling applications tend to tacitly condone knowledge or data gaps in order to come up with numbers. I often heard statements such as "we actually don't know, but this our best estimate". I rarely heard "we actually don't know". I suspect that admitting ignorance is rarely considered a business opportunity: there is a demand for solid numbers, and there will always be someone willing to meet that demand. In order to illustrate this, I would like to exemplarily cite some lines from the Water Footprint Assement Manual (Hoekstra et al. 2011):

- p. 43: "In general it is always preferable to find local data pertaining to the crop field location. In many cases it is too laborious to collect location-specific data given the purpose of the assessment."

- p. 44: "When applying the 'irrigation schedule option' in the CROPWAT model, one needs soil data; if no soil data are available we advise to choose 'medium soil' as a default."

- p. 81: "When data for a specific catchment are lacking we recommend to reckon with a default value of at least 12 per cent [of land reserved for conservation]."

- p. 119: "A major challenge is therefore to develop more detailed guidelines regarding what default data can be used when accurate local estimates are not available. In this context it is relevant to develop a database with default water footprint estimates for a large variety of processes and products, differentiating between production regions (such as countries). This would be very helpful for assessing the water footprints of consumers or producers, who know what they

buy but often do not know all relevant details on the production and supply chain of the things they buy."

I am aware that these citations are not directly pertinent to the issue of the freshwater PB, and admittedly, they are presented in a poignant way. However, they should illustrate how easy it is to generate numbers from "default" assumptions. And while most practitioners and scientists (including myself) will be familiar with the issue of making "rough assumptions", our responsibility is to transparently communicate the role of these assumptions with regard to our results.

In this context, Dr Lorenz is also right with his call for a more effective communication of limitations in hydrological and hydrometeorological monitoring and modelling information at different scales and for different purposes. However, the present case and many others demonstrate how difficult it is, even if the limitations *were* clearly stated, to prevent unwarranted applications.

*References:*

Hoekstra, A. Y., Chapagain, A. K., Aldaya, M. M. and Mekonnen, M. M.: The Water Footprint Assessment Manual - Setting the Global Standard, Earthscan, London, Washington D.C.

––––––––––––––––––––––––––

---

## Short Comment (SC3) · 25 Mar 2017

May I suggest the author a look to the current debate on the freshwater planetary boundary, very related to this discussion and which might be of his interest?

Jaramillo, F., and G. Destouni (2015), Comment on "Planetary boundaries: Guiding human development on a changing planet," Science, 348(6240), 1217–1217, doi:10.1126/science.aaa9629.

Gerten, D., J. Rockström, J. Heinke, W. Steffen, K. Richardson, and S. Cornell (2015), Response to Comment on "Planetary boundaries: Guiding human development on a changing planet," Science, 348(6240), 1217–1217, doi:10.1126/science.aab0031.

Cheers, Fernando

---

## Referee Comment (RC2) · HHG Savenije (Referee) · 26 Mar 2017

The advantage of starting late in the discussion is that one can benefit from the exchanges that have already taken place between author and reviewers. With regular papers, it is maybe not desirable to read the other reviews before writing one's own, but in the discussion of an opinion paper, such interaction is very valuable, I think.

First of all, I would like to complement the author on a well-argued opinion. The author makes a convincing case when he states that there is not really a planetary boundary for water use. For a regime shift to take place, there should be a clear non-linear feedback mechanism that can shift the system to another (non-desirable) system state; for instance large-scale desertification, or maybe large-scale flooding. The author fails to see such a mechanism, particularly not when we merely consider that water is managed within river basins. And why would the poor management in one river basin have global impact, or lead to some domino effect that tips the scale? I can follow this argument, but then the author misses an important feedback mechanism.

The global water resources are fed by moisture that is transported from the oceans to the land through the atmosphere. Indeed this moisture flux is not likely to be affected by our water use, although people like Makarieva and Gorskov (2007) (http://www.hydrol-earth-syst-sci.net/11/1013/2007/) state otherwise (and they may have a point). But more important is that this flux, as it travels over land, is influenced by our land use. Precipitation drains the flux, but evaporation from land (seen by many as a loss of water, which it is not) replenishes the moisture content of the atmosphere and sustains rainfall occurring downwind. Just like in the economy where economic activity has a multiplier effect on the GDP, the evaporation has a multiplier effect on the terrestrial precipitation. As was shown by Van der Ent et al. (2010), large parts of the world rely for 80% or more on moisture that has been recycled (sometimes several times) by evaporation from land. Most of China, large parts of Africa and South America rely for more than 80% on recycled moisture. Land use change thus has an impact that goes beyond the river basin boundary and has a global impact, particularly if such land-use change is the result of a policy to turn forest into crop land, or to overstock marginal grass lands. Although this is not the same as merely water use, it is clearly a change in water using activity that could have irreversible effects at global scale. And to strengthen the argument, Van der Ent et al. (2010) merely looked at the moisture content in the atmosphere. They did not consider the change in atmospheric circulation patterns that could result from changes in the energy budget associated to land use change

So I agree that it is hard to see a planetary boundary in water consumption itself, as long as this negative feedback mechanism is missing. And also that it is too simple to just put a cap on water use. But land use change definitely can have a negative feedback on terrestrial precipitation, and the impact of local changes can be much

more far reaching than is suggested purely by water withdrawal.

As an aside, I don't agree with anonymous referee#1, who found the author's language too colloquial. I am not at all sure if more formal language would help to make the message clearer. Please feel free to use the clear and well readable language used in this article. Particularly in an opinion paper such language is welcomed.

References:

Van der Ent, R.J., Savenije, H.H.G., Schaefli, B., Steele-Dunne, S. C., 2010. Origin and fate of atmospheric moisture over continents, Water Resour. Res. 46, W09525, 12pp. doi:10.1029/2010WR009127.

Makarieva, A. M. and Gorshkov, V. G.: Biotic pump of atmospheric moisture as driver of the hydrological cycle on land, Hydrol. Earth Syst. Sci., 11, 1013-1033, doi:10.5194/hess-11-1013-2007, 2007.

---

## Author Comment (AC4) · 27 Mar 2017

I would like to thank Dr Jaramillo for pointing out the commentary article by Jaramillo and Destouni (2015).

In that commentary, Jaramillo and Destouni argued that Steffen et al. (2015) *"ignored recent scientific advances implying that the global consumptive use of freshwater may have already crossed the associated planetary boundary"*, and presented studies which suggest a consumptive water use that is substantially higher than the 2600 km$^3$ figure published by Steffen et al. (2015). One key argument of the commentary was the finding that reservoirs can *"increase the net total actual evapotranspiration from the hydrological basins that include the reservoirs"*.

Please note that in the opinion paper, I referred, on p. 3, ll. 2-3, to Destouni et al.

(2013) instead of Jaramillo and Destouni (2015): *"Critical commentaries, however, mostly called for a revision of the actual numbers (e.g. Molden, 2009; Destouni et al., 2013)."*

I did this because the 2013 reference forms the original and *main* (though not exclusive) basis of the 2015 commentary, and already contains the statement that *"a global indication of the regional results is a net total increase of evapotranspiration that is larger than a proposed associated planetary boundary"*. In order not to overrepresent that position of the group around Jaramillo and Destouni, I did deliberately not include the 2015 reference on top. I also put an "e.g." before the citations on p. 3, ll. 2-3, in order to indicate that the corresponding references are not considered exhaustive.

In hindsight, however, I think it might be better to include the Jaramillo and Destouni (2015) reference into that statement in order to avoid the impression that the opinion paper misses a specific thread of the PB debate (as it occured to Dr Jaramillo). I also admit that Jaramillo and Destouni (2015) considered further studies and findings that had not been included with Destouni et al. (2013). And finally, I appreciate very much Jaramillo and Destouni's criticism regarding the *"message of apparent calm for freshwater"* which is being implied by Steffen et al. (2015). For those reasons, I think it is warranted to include the Jaramillo and Destouni (2015) reference in a revised version of the paper.

However, I have to emphasize that neither Destouni et al. (2013) nor Jaramillo and Destouni (2015) question the general concept of a planetary boundary on freshwater use. They only challenge the estimate of current human freshwater consumption. Accordingly, Gerten et al. (2015), in defence of Steffen et al. (2015), gratefully note *"that Jaramillo and Destouni accept the position of the freshwater planetary boundary but question our quantitative estimate of current human freshwater consumption."*

My opinion paper fundamentally challenges the concept of a planetary freshwater PB. Therefore, I want to avoid overemphasizing the debate on quantitative estimates. In

that regard, please also see my response to the short comment by Christof Lorenz (SC2) in this interactive discussion.

*References:*

Destouni, G., Jaramillo, F. and Prieto, C.: Hydroclimatic shifts driven by human water use for food and energy production, Nat. Clim. Change, 3, 213-217, 2013.

Gerten, D., J. Rockström, J. Heinke, W. Steffen, K. Richardson, and S. Cornell, Response to Comment on "Planetary boundaries: Guiding human development on a changing planet," Science, 348(6240), 1217–1217, 2015.

Jaramillo, F., and G. Destouni, Comment on "Planetary boundaries: Guiding human development on a changing planet," Science, 348(6240), 1217–1217, 2015

Molden D.: Planetary boundaries: The devil is in the detail, Nature Reports Climate Change, 3, 116-117, doi: 10.1038/climate.2009.97, 2009.

Steffen W., Richardson, K., Rockström, J., Cornell, S. E., Fetzer, I., Bennett, E. M., Biggs, R., Carpenter, S. R., de Vries, W., de Wit, C. A., Folke, C., Gerten, D., Heinke, J., Mace, G. M., Persson, L. M., Ramanathan, V., Reyers B. and Sörlin, S.: Planetary boundaries: Guiding human development on a changing planet, Science, 347, 1259855, doi: 10.1126/science.1259855, 2015.

---

## Referee Comment (RC3) · D. Gerten (Referee) · 9 Apr 2017

General

This paper suggests that hydrologists should not only disapprove but actively refute the concept of a "planetary boundary" (PB) for human freshwater use. The main reasoning is that there is no proof that local/regional transgressions of tolerable water use limits would lead to a large-scale (planetary) impact; that the current status of this PB wrongly indicates that the water situation is globally fine; and that it might be dangerous if politicians, companies etc. adopted the immature if not erroneous PB concept (for e.g. falsely suggesting that excessive water use in some locations could be offset through water savings elsewhere). I acknowledge that this is an opinion paper speaking for itself, but I like to rectify some points and refer to some recent developments that

the author should reflect (in addition to the points raised by previous commentators), as otherwise the concept of PBs and also the concept of water footprints is partly misrepresented.

Main points

It is true that the current approach to determine a PB for human freshwater use does not convey quantitative evidence that regional regime shifts in hydrological systems "could push the entire earth system away from its current state". Rather, as the author correctly notes, such massive hydrological changes need to be seen in the context of climate and land use changes. Actually, these interactions are reflected in the PB concept which clearly states that all 9 PBs are closely coupled – they altogether constitute the earth system and no single dimension should be considered in isolation. Also, the concept acknowledges that while single regional regime shifts or other hydrological disturbances may not induce larger-scale effects, but that occurrence of such events in many places might well do so (whether and where is not (yet) known, hence the positioning of the PB at the lower end of the scientific uncertainty zone).

That said, the PB concept is an effort to describe the status of the earth system as a whole as influenced by collective human activities, along its (currently nine) dimensions – of which the freshwater cycle is one. Indeed it is arguable to use the global consumptive blue water use as a proxy (i.e. as an integral variable) to capture the very complex ways by which humans alter the global water cycle; but instead of defeating the concept it can also be argued that we need better metrics to describe the global hydrological impact of human activities, ideally capturing how these interact with anthropogenic climate and land cover change. I also agree that the current notion that the PB for water use is still in a globally "safe space" (<4000 km3/yr) might provoke misunderstandings – but that is exactly the reason why in the Steffen et al. (2015) paper sub-global boundaries were introduced: these still cover only part of the human interferences with the water cycle (based on transgression of local environmental flow requirements) – but importantly they exhibit the very spatial pattern of detrimental

freshwater system alterations and water scarcity. Since the publications of Steffen et al. (2015) and also Gerten et al. (2013), this spatial information should no longer be separated from the global PB value, even if it is not yet satisfactorily solved how to adequately add up the regional transgressions in many places to a global value (as the author correctly notes).

In other words, the PB concept is in no way contradictory to the many studies on regional water problems – it is rather an attempt to integrate these into the larger picture of other global environmental problems, and yes, to detect whether and where there is a point beyond which the global situation may no longer be "acceptable" (certainly an ethical question too, which is why also ethicists try to develop the concept further, see Ziegler et al. 2017). The question here is: if the manifold and in many regions devastating human interferences with the water cycle are to be considered a part of the overall anthropogenic impact upon the earth system – and I am pretty sure they should, otherwise the role of water would be gravely downplayed! – how can we do that? Hence, the author might ask himself whether he would really like to give up asking this question, and why not rather invite hydrologists to provide convincing alternatives should they disagree with the current PB approach. Is it ever more maps of water scarcity, curve plots of increasing number of dams or aquatic species loss, or more detailed descriptions of local water problems (neglecting that they have more and more global drivers and impacts eventually requiring supraregional governance)? How to respond to the question whether and to what degree water problems are a global phenomenon, whether there is a "global water crisis"?

The usefulness of the PB concept lies in the fact that it puts forward the idea that the earth system is under severe pressure by humans along multiple dimensions, freshwater being one of them. I personally disagree with the author that its increasing recognition is a development of concern – because it rather increases awareness among policy-makers and business people that integrated and increasingly global perspectives on environmental issues including water issues are needed. There is no doubt

that many of its aspects and quantitative approaches are still premature, but science goes on to improve knowledge further, ideally in a co-creative manner with stakeholders as demonstrated by the transdisciplinary conference mentioned by the author and by various papers on the topic (e.g. Clift et al. 2017). While it can never be ruled out that some people try to abuse such concepts for their own purpose (as suggested by the discussion surrounding Fig. 2 in the opinion paper), most others (according to my experience) critically reflect the idea of a PB (for freshwater use) but at the same time think about how to develop it further qualitatively and quantitatively – in constructive ways.

The author also addresses aspects of the water footprint concept – I'm not sure what this criticism is actually about (in addition to what authors such as D. Wichelns have argued before) and what it tells us that A. Hoekstra and J. Rockström have not yet published papers together. But I would like to point to recent research that tries to unify the two concepts, also sorting out some misconceptions inherent to this opinion paper: Hoekstra & Wiedmann (2014), Fang et al. (2014), Fang et al. (2015), Hoekstra (2017). In this context I would like to highlight that the PB concept is about critical environmental limits to water use while the water footprint concept is about the actual magnitudes and locations of that use – they complement each other and should not be confused. And, if correctly understood, they both do not at all invite to "globally offset water-related environmental impacts" and they do not at all neglect the message that "the impacts of water scarcity on human welfare are already obvious".

Technical/minor points: Abstract and Section 1 are largely redundant.

References Fang, K., Heijungs, R., de Snoo, G.R. 2015a: Understanding the complementary linkages between environmental footprints and planetary boundaries in a footprint-boundary environmental sustainability assessment framework. Ecological Economics 114, 218-226. Fang, K., Heijungs, R., Duan, Z., de Snoo, G.R. 2015b: The environmental sustainability of nations: benchmarking the carbon, water and land footprints against allocated planetary boundaries. Gerten, D., Hoff, H., Rockström, J.,

Jägermeyr, J., Kummu, M., Pastor, A.V. 2013: Towards a revised planetary boundary for consumptive freshwater use: role of environmental flow requirements. COSUST 5, 551-558. Hoekstra, A.Y., Wiedmann, T.O. 2014: Humanity's unsustainable environmental footprint. Science 344, 1114-1117. Hoekstra, A.Y. 2017: Water footprint assessment: evolvement of a new research field. Water Resources and Management, doi:10.1007/s11269-017-1618-5. Clift, R., Sim, S., King, H. et al. 2017: The challenges of applying planetary boundaries as a basis for strategic decision-making in companies with global supply chains. Sustainability 9(2), 279. Steffen, W., Richardson, K., Rockström, J. et al. 2015: Planetary boundaries: guiding human development on a changing planet. Science 347, 1259855. Ziegler, R., Gerten, D., Döll, P. 2017: Safe, just and sufficient space – the planetary boundary for human water use in a more-than-human world. In: Ziegler, R. & Groenfeldt, D. (Eds.), Global Water Ethics – Towards a Global Ethics Charter, 109–130. Earthscan.

---

## Short Comment (SC5) · 2 May 2017

I appreciate the well articulated planetary boundary critique by Maik Heistermann. The scientific quest for improved definitions and quantifications of the planetary boundaries is a continuous challenge, and constructive critique is key to move the knowledge frontier forward.

Clearing two fundamental misunderstandings

This said, I find it important to clear out misunderstandings that lead to unsubstantiated critique, or worse, is at risk of dismissing important scientific questions.

Two fundamental misconceptions are clearly driving the arguments in Maik's paper. The first relates to what planetary boundaries (PB) are. The planetary boundaries de-

fine the environmental processes and systems that regulate the stability and resilience (the state) of the Earth system (with the inter-glacial Holocene state as the reference state). Earth system science over the past 30 years (driven not least by the Global Environmental Change programs, IGBP, WCRP, IHDP, Diversitas and ESSP including the Global Water System Program, GWSP) clearly shows that the Earth system is a complex self-regulating system where the hydrosphere interacts with the biosphere, atmosphere, cryosphere, geosphere and stratosphere. The global cycles of carbon, nitrogen, phosphorus, and water, interact across scales through the biosphere (oceans and land), atmosphere and stratosphere, to regulate the state of the Earth system. The fundamental role of planetary boundaries is thus to offer an integrated Earth system framework for defining the Earth system processes that regulate the state of the Earth system. Science shows clearly that water forms part of the fundamental fabric of the Earth system, functioning (as Prof. M. Falkenmark has pointed out) as "the bloodstream of the biosphere", through moisture feedbacks (and climate regulation), wettening of landscapes (retaining moisture levels and blue water stocks and flows regulating nutrient and carbon flows and stocks), biomass growth, biodiversity composition, and energy dynamics in the land-ocean interface (Steffen et al., 2004; Bhaduri et al., 2014).

Does this mean that every PB process/system that regulates the state of the Earth system needs to have planetary scale tipping points? Of course not. But Maik wrongly interprets the planetary boundaries framework as if this is the case, despite the fact that this is spelled out time and time again in both 2009 (Rockström et al) and 2015 (Steffen et al) planetary boundaries' papers. As shown in Fig 1. (Steffen et al., 2015), two types of processes/systems qualify as planetary boundaries; (1) those where we have scientific evidence of planetary scale thresholds (the boundaries for the climate system, ozone depletion) and (2) those where we do not (currently) have evidence of planetary scale tipping points but which are processes operating across scales to regulate the direction of Earth system feedbacks (negative or positive). These "slow variables" are identified as the boundaries for water, land, biodiversity and biogeochemical flows (N

and P boundary). The point is thus that these boundaries (the slow variables), despite not having clear evidence of planetary scale tipping points (some scholars do question this, e.g., Barnosky et al. 2011, indicating that there very likely is a planetary tipping point for biodiversity), contribute to regulate the state of the Earth system.

The water boundary is a clear case in point. While we do not have evidence of planetary scale water tipping points, there is clear evidence that water contributes to regulate the state of the biosphere and that the global hydrological cycle has entered the Anthropocene, i.e., that human shifts in water flows across scales can affect the functioning of Earth system processes, again, across scales (Vörösmarty et al., 2013). The water boundary is defined in this way. In the 2015 update we proposed a dual definition of the water boundary to reflect the fact that water operates at ecosystem/watershed/basin scale. The key question is, what is the minimum level of wettening (green and blue stocks/flows) of landscapes/river basins required to maintain the functioning of biomes and land systems? Admittedly, as Maik points out, this is a challenging research question. It is a scientific question to provide evidence, for river basins across the Earth system, of the minimum amount of freshwater required - across time and space - to maintain critical Earth system functions intact (e.g., carbon sinks and moisture feedback).

We have chosen, as a proxy, two control variables as our proposed scientific indicators for the water boundary - at the global scale the maximum cumulative volume of consumptive blue water, and at the river basin scale, the minimum volumes of environmental water flow (EWF) that are required for ecosystem stability.

Maik's misconception leads to some quite harsh and unsubstantiated statements, arguing, e.g., that the planetary boundary concept "suggests that we can globally offset water-related environmental impacts, a notion that defies both common sense and hydrological science." However, we (Steffen et al., 2015) repeatedly describe freshwater as a regional process, making it clear that "sub-global dynamics are critical" for freshwater use, and that freshwater use is a PB with "strong regional operating scales". We

furthermore state clearly that "we emphasize that our sub-global-level focus is based on the necessity to consider this level to understand the functioning of the Earth System as a whole. The PB framework is therefore meant to complement, not replace or supersede, efforts to address local and regional environmental issues." Nowhere do we suggest that water related environmental impacts can be "globally offset". The differentiated response of ecosystems and climate to hydrological changes as a function of space and time is fully acknowledged. In this case it appears Maik is trying to create an intellectual debate over opposing views that do not exist.

And here comes misunderstanding No 2. We did not choose "consumptive blue water use" as a control variable for the aggregate global scale water boundary, as an estimated of allowed "maximum human water use". We are only interested in the maximum allowed interference with the hydrological cycle, before which we risk seeing non-linear shifts in biome and river basin functioning, which in turn may trigger feedbacks affecting - across scales - the stability of the Earth system. We concluded that the net cumulative and consumptive reductions in runoff water from the global hydrological, is the best parameter we have (so far) in the hydrological cycle, to reflect the stability of water functions in the biosphere. So, it has nothing to do with human water use. It has to do with the maximum level of shifts in the global hydrological cycle beyond which we are likely to see changes in feedbacks, potentially triggering non-linear shifts in other Earth system regulation processes (e.g., carbon sinks, biodiversity, moisture feedback).

Why "consumptive use of blue water"? Well, because it is a good indicator variable of the final end point of all the changes/dynamics that occur in (i) the partitioning of water in the hydrological cycle, and (ii) the flow of water through landscapes. For example, if rainfall (P) shifts, and/or green water flows as (ET) shifts in the 1st partitioning point in the water balance (e.g., through land use change/increased green water use), this will affect the volumes of surface and sub-surface runoff (R), which in turn are affected by withdrawals of R, generating a net final impact (degree of drying) in river basins, but only after considering "consumptive use", i.e., factoring in return flow of runoff. So,

selecting consumptive use of blue water, is solely to find the best possible parameter to reflect the degree of change in the hydrological cycle, i.e., shifts in blue runoff flow to green vapour flows, e.g., changing moisture feedback, which, as Huub Savenije shows clearly in his commentary, has strong evidence not only of affecting rainfall, but also having tele-connections across biomes/regions, i.e., function as a planetary boundary.

Are there water thresholds?

A fundamental reason for Maik's critique appears to be his questioning whether there is evidence of water induced tipping points (shifts in feedbacks triggering non-linear dynamics and ultimately state shifts).

To start with, here Maik appears to apply very narrow criteria for scientific evidence of non-linear water dynamics to be accepted as relevant. He seems to imply that it is only non-linear dynamics caused by human water use that is relevant. This is of course incorrect. Just like in climate science, where paleo-climatic data on sea-level rise estimates from the Eemian (last time we had +2 C) is relevant for us to understand what "human climate interference" implies today, hydrological science evidence showing what happens with biomes when water flows shifts, is relevant, irrespective if it is us humans or natural variability causing it!

The purpose of the PB framework is to quantify the role of water in sustaining the stability of the Earth system, irrespective of whether changes in the hydrological cycle are caused by natural variability or human interference.

There is ample evidence of water induced tipping points in ecosystems, references made not least in Rockström et al. (2009) and Steffen et al. (2015). Maik makes a point of lifting Brooks et al. (2013) to the forth, which was an attempt of questioning the core notion of whether there are feedback shifts and biophysical thresholds in the Earth system. What Maik chooses to omit though is the comprehensive response by Hughes et al., (2013), convincingly dismissing Brooks et al's rather unsubstantiated critique, with ample references to the scientific literature.
It is superfluous to run through all the evidence of non-linear water induced dynamics in the Earth system (at different scales) in this commentary. It suffices to refer to papers on water induced transitions in forest to savanna states (Staver et al., 2011; Horita et al., 2011), water related teleconnections across biomes (Snyder et al., 2010), water induced desertification in biomes (Scheffer and Carpenter 2003), major basin transitions such as the Aral sea and lake Chad (Rockström et al., 2014), and critical state transitions in ecosystems related to water (Carpenter et al., 2015).

Human water use affecting feedbacks beyond river basin scale

Furthermore, there is significant research showing that decrease in runoff (caused by increase in consumptive water use), can reduce rainfall, i.e., that landscape "wettening" matters for moisture feedbacks and thus rainfall generation. For example, irrigation can destabilize the monsoon (Guimberteau et al., 2011; Im et al., 2013; Lee et al., 2009; Tuinenburg et al., 2014).

There is a red thread here, where Maik purposefully seems to ignore evidence that shows that interference with the water balance and hydrological cycle can generate [water related] climatic feedback and interactions with other processes. Climate model simulations do show that land surface response can amplify monsoonal shifts – e.g., leading to large scale tipping between an arid and green state of the Sahara (e.g., Tierney et al. 2017).

Finally, on why it is good to have a dual water boundary on both global interference with the hydrological cycle, and maintaining minimum levels of wettening in biomes (as moisture and runoff). In the Anthropocene, with influence across all river basins, we can think of a situation where humans perturb the climate system through water use even if river flows are largely retained. Make the thought experiment that blue water is harvested and released as evaporation at the land-ocean interface everywhere on the planet [this is consumptive water use]. This will in a first step not decrease net flows in blue water stocks, but will add an immense amount of water vapour and latent heat

flux to the atmosphere with potential implications for the climate system.

Water - a planetary boundary

Another thought experiment, to elucidate the "proof" of water being a planetary boundary, is to shut down the global hydrological cycle. Clearly, we would then have a fundamental state-shift of the planet, with collapse of the climate system and biosphere. Gradually adding water back to the system, and at a certain point we are bound to see the Earth system kick-in its biogeochemical processes enabling living conditions (as we know them) back on Earth. This thought experiment clearly shows that somewhere - along the water flow line - there is a planetary boundary on water. It is complex, operating across scales, intimately connected with land, ocean and climate. It is a grand scientific quest to continue working towards understanding and defining the safe operating space for water in the Anthropocene. My advice to Maik is therefore, to think more holistically of the Earth System, and not solely focus on incremental decrease/increase of water flows.

References:

Barnosky, A.D., et al. 2011. Has the Earth's sixth mass extinction already arrived? Nature 471, 51-57 (2011). doi:10.1038/nature09678

Bhaduri, A., Bogardi, J., Leentvaar, J., Marx, S., (Eds.), 2014. The Global Water System in the Anthropocene. Challenges for Science and Governance. Springer

Brook, B.W. et al. (2013) Does the terrestrial biosphere have planetary tipping points? Trends Ecol. Evol. 28, 396–401Âǎ

Carpenter, S.R., Brock, W.A., Folke, C., van Nes, E.H., and Scheffer, M., 2015. Allowing variance may enlarge the safe operating space for exploited ecosystems. PNAS, 112 (45) : 14384 - 14389

Guimberteau, M., Laval, K., Perrier, A. and Polcher, J.: Global effect of irrigation and its impact on the onset of the Indian summer monsoon, Clim. Dyn., 39(6), 1329–1348,

doi:10.1007/s00382-011-1252-5, 2011.

Horita, M., Holmgren, M., Van Nes, E.H., and Scheffer, M., 2011. Global Resilience of Tropical Forest and Savanna to Critical Transitions

Hughes, T.P., S. Carpenter, J. Rockström, M. Scheffer, B. Walker, Multiscale regime shifts and planetary boundaries. Trends Ecol. & Evol. 28, 389-395 (2013). doi:10.1016/j.tree.2013.05.019

Im, E.-S., Marcella, M. P. and Eltahir, E. A. B.: Impact of potential large-scale irrigation on the West African Monsoon and its dependence on location of irrigated area, J. Clim., 131009131322009, doi:10.1175/JCLI-D-13-00290.1, 2013.

Lee, E., Chase, T. N., Rajagopalan, B., Barry, R. G., Biggs, T. W. and Lawrence, P. J.: Effects of irrigation and vegetation activity on early Indian summer monsoon variability, Int. J. Climatol., 29(4), 573–581, doi:10.1002/joc.1721, 2009.

Rockström, J. et al.., 2009. A Safe Operating Space for Humanity. Nature, 461: 472 - 475

Rockström et al., 2014. Water Resilience for Human Prosperity. Cambridge University Press

Steffen et al., 2004. Global Change and the Earth System. Springer-Verlag Berlin Heidelberg New York. ISBN 3-540-40800-2.

Steffen, W., Richardson, K., Rockström, J., et al., 2015. Planetary Boundaries: Guiding Human Development on a Changing Planet. Science, 347 (6223): DOI: 10.1126/science.1259855

Snyder, P.K. (2010) The influence of tropical deforestation on the Northern Hemisphere climate by atmospheric teleconnections. Earth Interact. 14, 1–34Âă

Staver, A.C. et al., 2011. Alternative Biome States. Science, 334: 230-232

Scheffer, M., and Carpenter, S., 2003. Catastrophic regime shifts in ecosystems: linking theory to observation. TRENDS in Ecology and Evolution, 18 (12) : 648-656

Tierney, J. E., Pausata, F. S. R. and deMenocal, P. B.: Rainfall regimes of the Green Sahara, Sci. Adv., 3(1), 2017.

Tuinenburg, O. A., Hutjes, R. W. A., Stacke, T., Wiltshire, A. and Lucas-Picher, P.: Effects of Irrigation in India on the Atmospheric Water Budget, J. Hydrometeorol., 15(3), 1028–1050, doi:10.1175/JHM-D-13-078.1, 2014.

Vörösmarthy, C.J., Pahl-Wostl, Claudia, Bhaduri, A., 2013. Water in the anthropocene: New perspectives for global sustainability. Current Opinion in Environmental Sustainability 2013, 5:535–538Âă

---

## Author Comment (AC8) · 5 May 2017

I would like to thank Prof. Rockström for his timely contribution. I will try to respond without repeating too much of what has already been said in the two months since the discussion had started (some of Prof. Rockström's points have already been addressed in my response to RC3).

When I originally wrote the manuscript of the opinion paper, I was indeed kind of concerned. To me, the idea of a planetary boundary on freshwater use appeared so obviously flawed that I suspected I had missed any important aspects. Since then, people (including Prof. Rockström himself, in his comment) have repeatedly been suggesting that I "misunderstood", "misconceived", or "misinterpreted" the concept.

At this point in the interactive discussion, I am confident that this is not the case. That

confidence is based more on the attempts to argue *in favour* of the PB concept than on the comments that actually shared my reservations. What I'd agree to, however, is that the concept of the freshwater PB in fact holds quite some potential for misunderstanding.

Altogether, Prof. Rockström points out "two fundamental misunderstandings" in the opinion paper and the subsequent interactive discussion.

**1st misunderstanding**

Prof. Rockström reiterates that "the planetary boundaries define the environmental processes and systems that regulate the stability and resilience [...] of the Earth system", and establishes that the water cycle in fact contributes massively to regulate the state of the Earth system (pp. C2-C3). Who would oppose?

The alleged first misconception, as I understand Prof. Rockström, is that I "wrongly interpreted the planetary boundaries framework as if [there were a planetary tipping point for freshwater]", although the PB literature never claimed such a planetary scale tipping point. In fact, that notion was repeatedly emphasized in the PB literature - and it was also explicitely voiced in the opinion paper (p. 3, II. 13-17). What remains, however, after emphasizing that the water cycle is "operating across scales", is a planetary boundary on freshwater use, and the unambiguous statement that "transgressing one or more planetary boundaries may be deleterious or even catastrophic due to the risk of crossing thresholds that will trigger non-linear, abrupt environmental change within continental- to planetary-scale systems" and that "each proposed boundary position assumes that no other boundaries are transgressed" (Rockström et al. 2009). The inconsistency between these two notions ("planetary boundary, but no planetary tipping point") is symptomatic for the concepts behind the freshwater PB: the "duality" of planetary and basin-scale boundaries, or the idea of having distinct boundaries that should, however, not be viewed in isolation (see also RC3). In my opinion, the common ground of these inconsistencies is a lack of coherent scientific underpinning.

**2nd misunderstanding**

As for the alleged second misunderstanding, I have to admit that I found the related paragraphs of Prof. Rockström's comment a bit difficult to follow. So before trying to rephrase, I will quote both paragraphs in full (pp. C4-C5):

"And here comes misunderstanding No 2. We did not choose 'consumptive blue water use' as a control variable for the aggregate global scale water boundary, as an estimate of allowed 'maximum human water use'. We are only interested in the maximum allowed interference with the hydrological cycle, before which we risk seeing non-linear shifts in biome and river basin functioning, which in turn may trigger feedbacks affecting - across scales - the stability of the Earth system. We concluded that the net cumulative and consumptive reductions in runoff water from the global hydrological, is the best parameter we have (so far) in the hydrological cycle, to reflect the stability of water functions in the biosphere. So, it has nothing to do with human water use. It has to do with the maximum level of shifts in the global hydrological cycle beyond which we are likely to see changes in feedbacks, potentially triggering non-linear shifts in other Earth system regulation processes (e.g., carbon sinks, biodiversity, moisture feedback).

Why 'consumptive use of blue water'? Well, because it is a good indicator variable of the final end point of all the changes/dynamics that occur in (i) the partitioning of water in the hydrological cycle, and (ii) the flow of water through landscapes. For example, if rainfall (P) shifts, and/or green water flows as (ET) shifts in the 1st partitioning point in the water balance (e.g., through land use change/increased green water use), this will affect the volumes of surface and sub-surface runoff (R), which in turn are affected by withdrawals of R, generating a net final impact (degree of drying) in river basins, but only after considering 'consumptive use', i.e., factoring in return flow of runoff. So, selecting consumptive use of blue water, is solely to find the best possible parameter to reflect the degree of change in the hydrological cycle, i.e., shifts in blue runoff flow to green vapour flows, e.g., changing moisture feedback, which, as Huub Savenije shows clearly in his commentary, has strong evidence not only of affecting rainfall, but also having tele-connections across biomes/regions, i.e., function as a planetary boundary."

As far as I understand, Prof. Rockström tries to make a point that human interference with the water cycle should not be reduced to direct consumptive use of freshwater (e.g. by irrigated agriculture). Instead, he claims that the control variable of the freshwater PB should be interpreted as *"the net cumulative and consumptive reductions in runoff"* (I suppose this implies "natural runoff", although the PB literature lacks the clear definiton of a "natural" reference), and implies that these net effects should also include impacts of e.g. land use change on the net water balance. While this interpretation of "consumptive blue water use" appears a bit unusual, it is consistent with the PB literature insofar as it lumps together different processes with different signs.

Prof. Rockström concludes that "the purpose of the PB framework is to quantify the role of water in sustaining the stability of the Earth system, irrespective of whether changes in the hydrological cycle are caused by natural variability or human interference". He (re)defines the planetary freshwater boundary to be about "water" and its "role" - in a more general sense. This is quite convenient since the supposed "ample evidence of water induced tipping points" quoted by the PB literature is utterly unrelated to human freshwater consumption. But while, according to Prof. Rockström, "the freshwater PB has nothing to do with human water use" (!), it is still referred to, in the PB literature, as the planetary boundary on freshwater use, with consumptive blue water use as the control variable. At this point, I am finally lost for words.

**What now?**

I think the essence of Prof. Rockström's comment is that water is of paramount importance for the functioning of the Earth system, and that such importance must be reflected in a water-related planetary boundary. In my view, that is a fallacy. The water cycle is an inherent part of the climate system, and a key agent to mediate transport and conversion of matter and energy. The importance of water cannot be highlighted by putting it into an isolated PB. While PB proponents continue to argue that "the freshwater PB should not be viewed in isolation" (see RC3), it merely demonstrates that the boundary is ill-defined: neither does it highlight the functional relevance of water in the Earth system, nor does it demonstrate, as an indicator, the urgency of sustainable water resources management.

So does the water cycle contribute to "regulate the stability and resilience [...] of the Earth system"? Of course! Does the planetary boundary on freshwater use reflect that role of the water cycle in the Earth system? I don't think so, and it is convenient for me to disagree. This is because the burden of proof is not mine: to prove that "consumptive blue water use" or "global reduction of runoff" beyond whatever limit is an adequate proxy to indicate the risk of system collapse. If you argue, however, that water-induced changes will act regionally (or "accross scales"), you should understand and quantify the specific causal chains or loops.

Prof. Rockström gave me the advice *"to think more holistically of the Earth System"*. If I were to return the favour, my constructive advice would be to take a glance at the "Zen of Python" (Peters, 2004), an aphoristic collection of Pythonic design principles that, in my view, are surprisingly relevant to the matter, and which I hereby selectively quote:

[...]

Explicit is better than implicit.

Simple is better than complex.

Complex is better than complicated.

[...]

Readability counts.

Special cases aren't special enough to break the rules.

Although practicality beats purity.

Errors should never pass silently.

Unless explicitly silenced.

In the face of ambiguity, refuse the temptation to guess.

[...]

If the implementation is hard to explain, it's a bad idea.

If the implementation is easy to explain, it may be a good idea.

[...]

I would like to let these principles speak for themselves, and let the reader decide if he or she find's analogies to the issue under discussion. Just that much: "explicit is better than implicit" - example: if you think that human freshwater consumption affected terrestrial moisture recycling or atmospheric circulation patterns to a level that destabilizes the entire Earth system, then follow that thought and collect the evidence.

In fact, Prof. Rockström has provided a good starting point: after all, he presents four papers that demonstrate how irrigation might affect the climate system (Guimberteau et al., 2011; Im et al., 2014; Lee et al., 2009; Tuinenburg et al., 2014). Excellent!

These are the first references quoted in this discussion that establish a link between consumptive freshwater use (irrigation) and regional climate systems. Instead of accusing me to "purposefully ignore evidence", it might have been a good idea to include these references in the Steffen et al. (2015) update of the PB framework. Why was none of the these papers cited as evidence to support the existence or quantification of a freshwater PB? I will certainly not speculate about the reasons. These are all interesting papers, and they demonstrate that the functional relationships between irrigation and (monsoonal) rainfall patterns are highly complex. Guimberteau et al. (2011) compare "a 30-year simulation which takes into account irrigation [...] with a simulation which does not." They find that "differences are usually not significant on average over all land surfaces but hydrological variables are significantly affected by irrigation over some of the main irrigated river basins." Im et al. (2014), Lee et al. (2009) and Tuinenburg et al. (2014) find that the irrigation-induced cooling effect appears to be a dominant process as it tends to decrease atmospheric moisture convergence under specific constellations. All papers demonstrate that a decrease in precipitation in one region is typically tied to an increase in another region. What **none** of these papers demonstrates, yet, is a mechanism for the collapse of regional hydrological cycles, induced by consumptive water use. From that, we should neither draw the conclusion that such a collapse is impossible or inconceivable, nor should we start guessing numbers numbers at which level of water withdrawel or consumption that collapse might still occur. What we should do is to follow up on that research, and I encourage the PB community to do so. I thus vigorously reject Prof. Rockström's allegation that fundamentally challenging the concept of a freshwater PB created the "risk of dismissing important scientific questions". I merely suggest to replace speculation by a continued quest for scientific evidence.

While simulation experiments such as the ones discussed in the previous paragraph can be very useful to understand the mechanisms that govern system dynamics, I fail to see how the "thought experiments" provided by Prof. Rockström can be of help. Mental models (or thought experiments) are of limited value in understanding emergent

properties of non-linear systems.

And finally, about the controversial issue of "water offsetting" - as this has, for obvious reasons, raised some emotions: Prof. Rockström argues that "nowhere do we suggest that water related environmental impacts can be 'globally offset'". He is right. Neither Rockström et al. (2009) nor Steffen et al. (2015) suggested such an idea, and it is also fair to assume that it was not their intention to imply anything like it. Still, I insist that it is way too convenient to back out saying that such interpretation was unintended and that there is no way to prevent misuse of the concept. Why? Because a **planetary boundary on freshwater use** is a point-blank invitation to legitimate concepts such as "water-neutrality" or "water offsetting".

**Concluding remarks**

I have the feeling that the tone of the discussion has become increasingly bitter, and I honestly regret that. I guess the occasionally provocative style of the opinion paper has contributed its share, and an alleged "lack of evidence" might sound, in one scientist's ear, similarly harsh as "purposefully ignoring evidence" in another's. This is why I would like to use the opportunity, again, to particularly thank Prof. Gerten and Prof. Rockström for taking a stand to defend the freshwater PB. Those readers of HESS who had the perseverance to follow through the discussion might have learned a few things that they had not been aware of. That certainly holds true for myself, and I am grateful for that opportunity. While my position towards the freshwater PB is still crystal clear, I will certainly make use of these insights in a potential revision of the manuscript in order to make the message clearer and avoid misunderstandings.

**References**

Im, E.-S., Marcella, M. P. and Eltahir, E. A. B.: Impact of potential large-scale irrigation on theWest African Monsoon and its dependence on location of irrigated area, J. Clim., 131009131322009, doi:10.1175/JCLI-D-13-00290.1, 2013.

Lee, E., Chase, T. N., Rajagopalan, B., Barry, R. G., Biggs, T. W. and Lawrence, P. J.: Effects of irrigation and vegetation activity on early Indian summer monsoon variability, Int. J. Climatol., 29(4), 573–581, doi:10.1002/joc.1721, 2009.

Guimberteau, M., Laval, K., Perrier, A. and Polcher, J.: Global effect of irrigation and its impact on the onset of the Indian summer monsoon, Clim. Dyn., 39(6), 1329–1348, doi:10.1007/s00382-011-1252-5, 2011.

Peters, T. (2004): PEP 20 – The Zen of Python, URL: https://www.python.org/dev/peps/pep-0020/

Rockström, J., Steffen, W., Noone, K., Persson, A., Chapin, F. S., Lambin, E. F., Lenton, T. M., Scheffer, M., Folke, C., Schellnhuber, H. J., et al.: Planetary Boundaries: Exploring the Safe Operating Space for Humanity, Ecol. Soc., 14(2), 32-64, 2009.

Steffen W., Richardson, K., Rockström, J., Cornell, S. E., Fetzer, I., Bennett, E. M., Biggs, R., Carpenter, S. R., de Vries, W., de Wit, C. A., Folke, C., Gerten, D., Heinke, J., Mace, G. M., Persson, L. M., Ramanathan, V., Reyers B. and Sörlin, S.: Planetary boundaries: Guiding human development on a changing planet, Science, 347, 1259855, doi: 10.1126/science.1259855, 2015.

Tuinenburg, O. A., Hutjes, R. W. A., Stacke, T., Wiltshire, A. and Lucas-Picher, P.: Effects of Irrigation in India on the AtmosphericWater Budget, J. Hydrometeorol., 15(3), 1028–1050, doi:10.1175/JHM-D-13-078.1, 2014.

**C9**

---

## Author Comment (AC10) · 22 May 2017

In AC9, I suggested that any revision of the freshwater PB should be *explicit* about the processes which we think might be able to *"push the Earth system outside the stable environmental state of the Holocene"*, and also be *explicit* about our existing knowledge gaps. If, e.g., the disruption of terrestrial moisture recycling is considered crucial, then the boundary should be redefined accordingly, including an explicit acknowledgement of our fundamental gaps in qualitative and quantitative understanding. Fig. 1 of this addendum exemplifies (!) how such a revision might look like in terms of the main PB figure. To be sure, however, such a step would still not answer why regional (or "cross-scale") processes should be represented by a global number.

[Figure]

**Fig. 1.** (a) shows the original core figure of the PB framework; (b) exemplifies the idea of being "explicit" with regard to the processes and our existing knowledge gaps.